# Mate Choice Plurality, Choice Overload, and Singlehood: Are More Options Always Better?

**DOI:** 10.3390/bs14080703

**Published:** 2024-08-12

**Authors:** Menelaos Apostolou, Loukia Constantinidou, Antonios Kagialis

**Affiliations:** 1Department of Social Sciences, University of Nicosia, 46 Makedonitissas Ave., Nicosia 1700, Cyprus; constantinidou.l1@live.unic.ac.cy; 2Department of Psychiatry, School of Medicine, University of Crete, 710 03 Heraklion, Greece; kaganthony@gmail.com

**Keywords:** singlehood, choice overload, mate choice plurality, mating

## Abstract

A lack of options can make it challenging for individuals to find a desirable intimate partner. Conversely, an abundance of choices might lead to mate choice overload, making it difficult to determine the most suitable match. Additionally, having numerous alternatives after entering a relationship could undermine its stability by decreasing satisfaction with the current partner. The present research aimed to examine the effects of mate choice plurality on singlehood status and the willingness to stay in a relationship within the Greek cultural context. Specifically, we employed closed-ended questionnaires, which included instruments developed using AI, with a sample of 804 Greek-speaking participants. We found that participants who perceived they had a wider range of potential romantic partners reported a lower likelihood of being single. Furthermore, more perceived mate choices were associated with fewer years spent as single. However, mate choice plurality was also linked to higher choice overload, which, in turn, increased the likelihood of being single rather than in an intimate relationship. Moreover, more perceived mate choices were associated with more regrets about being in the current relationship. These regrets were linked to lower relationship satisfaction and ultimately contributed to a decreased willingness to stay in the relationship. Notably, this indirect effect was significant only for male participants.

## 1. Introduction

Being single—that is, not having an intimate partner—constitutes a common state in contemporary societies [1,2]. One reason people give for not having an intimate partner is that they are limited in their choices of desirable mates [3]. On the other hand, research indicates that having too many options may also prevent people from making any choice [4,5] or to make them dissatisfied with their current choices [6]. Accordingly, the purpose of the current research is to investigate the impact of perceived mate options on singlehood status and on a willingness to stay in a relationship. We will first argue that more choices are associated with an increased likelihood of finding an intimate partner and escaping singlehood. Subsequently, we will argue that too many choices may result in people facing difficulties in deciding who is right for them, a phenomenon known as choice overload, thereby increasing the probability of staying single. We will also argue that having many choices can decrease satisfaction with the current partner and increase one’s willingness to terminate the current relationship.

### 1.1. Mate Choice Plurality and Singlehood

The number of mate choices, or choice plurality, would be linked with singlehood status. To illustrate an extreme case, if individuals live in a context where no potential mates are available, they will inevitably remain single. Yet, even when potential mates are available, this does not guarantee that individuals will find a suitable match, as people are selective in their mate choices. More specifically, individuals do not choose mates randomly, but they have specific criteria in mind [7]. In particular, mate choice is guided by specific preferences, with individuals favoring potential partners who score highly in traits such as good looks, intelligence, and kindness [8]. They also desire their mates to match their own scores in these traits [9,10]. For example, they seek mates who are intelligent but not substantially more so than themselves [10].

High selectivity implies that individuals do not settle for the first potential mate they encounter, but rather screen prospective mates for desirable traits. As a result, the more options individuals have, the easier it will be for them to find a suitable match [11,12]. It follows that the number of available options will predict singlehood status: The fewer options individuals have, the more likely they are to be single (H1). Consistent with this prediction, when asked why they were single, individuals cited a lack of available options [3]. For instance, individuals living in small villages or working in environments predominantly consisting of the same sex, reported difficulties in meeting desirable mates [13]. Yet, having many options is not always beneficial for finding an intimate partner.

### 1.2. Choice Overload and Singlehood

When making choices, having options is generally beneficial as it increases the likelihood of finding what one desires. Yet, an abundance of choices can also have a negative impact, leading to choice overload. This occurs when individuals, overwhelmed by the number of available options, are unable to make a decision [4,5,14]. For instance, in a classic study on consumer choice [15], a tasting table with exotic jams was set up at an upscale grocery store. The table displayed either a small assortment of six jams or a large assortment of 24 jams. They found that 30% of consumers who saw the small assortment actually purchased a jam, whereas only 3% did so when presented with the large assortment. The researchers argued that their findings were a consequence of choice overload: Too many options decreased the motivation to make a choice. Similar results have been found in other studies on consumer choice involving different items, including pens [16], chocolates [17], gift boxes [18], and coffee [19].

Choice overload can also be present in the domain of mating. Online dating applications such as Tinder and Badoo provide access to a potentially large pool of prospective mates, making a choice difficult. For example, a daily user on Tinder is presented with an average of 140 partner options per day [20]. Research on online dating has found that access to more potential partner profiles resulted in more searching, more time spent evaluating less suitable options, a lower likelihood of selecting the best fit [21], and an increased likelihood of rejecting potential partners [22]. Accordingly, choice overload is an issue in the domain of mating: Individuals with many choices are likely to face difficulties in deciding who is best for them (H2). In turn, this can affect relationship status: Those who struggle to decide are more likely to remain single than those who do not face such difficulties (H3). These two predictions suggest an indirect effect of mate choice plurality on relationship status through choice overload: Individuals with many mate choices may become overwhelmed, face difficulties in deciding who is best for them, and consequently have a higher probability of remaining single (H4).

Furthermore, it has been argued that individuals experiencing choice overload are less likely to be satisfied with their decisions [23], less confident that they have chosen the best option [24], and more prone to post-decision regret [25]. Studies on online dating have found that as the choice of prospective partners increases, individuals become less satisfied with their partner choice and more prone to reverse this decision [26]. On this basis, we propose that there may be an additional pathway through which mate choice plurality leads to a higher incidence of singlehood. Specifically, individuals in a relationship who perceive themselves to have many mating options will be more likely to regret their current choice, leading to lower relationship satisfaction and a greater willingness to end their existing relationship (H5).

### 1.3. The Current Study

Existing literature on singlehood suggests that the availability of mate choices predicts relationship status. However, to the best of our knowledge, no study has attempted to examine this association. Furthermore, studies on online dating indicate that mate choice plurality may negatively impact the process of finding an intimate partner by causing choice overload. Yet, choice overload need not be confined to online dating. Individuals may find themselves with many choices due to possessing desirable traits (e.g., good looks), situational factors (e.g., living in a context with limited competition such as being a man in Germany following the end of World War II), or a combination of the two (e.g., being a good-looking man in Germany at the end of World War II). Therefore, this study aims to enrich existing literature by examining the association between mate choice plurality, choice overload, and singlehood. Additionally, mate choice plurality can negatively impact relationship stability, which can, in turn, affect relationship status. This association is another aspect that the present study aims to examine. In total, five hypotheses will be tested:

**H1:** 
*Mate choice plurality would be associated with relationship status: The fewer options individuals have, the more likely they are to be single.*


**H2:** *Mate choice plurality would be associated with choice overload: The more choices individuals have, the more likely they are to face difficulties in choosing between them*.

**H3:** *Choice overload would be associated with relationship status: The more difficulties individuals face in choosing between available mates, the more likely they are to remain single*.

**H4:** *Mate choice plurality would be indirectly associated with relationship status through choice overload: A perceived abundance of mate choices would be associated with increased choice overload, which would in turn, be associated with a higher probability of being single rather than in an intimate relationship*.

**H5:** *Mate choice plurality would be indirectly associated with willingness to stay in the relationship: Individuals in a relationship who perceive themselves to have many mating options would be more likely to regret their current choice, leading to lower relationship satisfaction and a greater willingness to end their existing relationship*.

We anticipate that hypotheses one, three, and four will also apply to the duration of singlehood. That is, mate choice plurality would be associated with fewer years of being single, choice overload with more years of being single, and a greater number of mating choices would be associated with more years of being single due to higher choice overload. Furthermore, existing literature indicates that different predictors of singlehood do not have the same impact on the relationship status of men and women [27]. Therefore, to examine whether this was the case for our variables of interest, we tested our hypotheses on the pooled sample, as well as separately on male and female participants.

Additionally, self-esteem, or one’s perceived worthiness as a person [28], is likely to be correlated with perceived mate choice plurality. Specifically, individuals with high self-esteem may perceive themselves to be more attractive as mates, and thus, have more mating options. Conversely, having more mating options may positively impact one’s self-esteem, as they may see themselves as more desirable mates. In essence, perceived mate choice plurality would correlate with self-esteem and could also act as a proxy for it. As self-esteem has a significant effect on singlehood [27], and to avoid any confounding effects, in the present study, we also measured participants’ self-esteem to keep it statistically constant. Furthermore, people with high self-esteem may be less satisfied with their present relationship and thus more willing to end it, as they may consider that they are worth more than their current partner gives them. Accordingly, self-esteem needs to be controlled when examining the effect of mate choice plurality on the willingness to continue an intimate relationship.

## 2. Materials and Methods

### 2.1. Participants

The study was conducted at a private university in the Republic of Cyprus, and received approval from the University’s Ethics Review Board. Participants were recruited in three different ways: (a) The study link was forwarded to students and colleagues with a request to share it within their social network; (b) the study was promoted on social media platforms (Facebook, Instagram), targeting participants residing in Greece and the Republic of Cyprus; and (c) two research assistants recruited participants in malls and coffee shops in the cities of Athens and Nicosia. For this purpose, a QR code was created, which led to our study and could be scanned by participants using their mobile phones.

In total, 804 Greek-speaking individuals participated in the study. In particular, the sample included 456 women, 335 men, eight participants who identified their sex as “other”, and five participants who did not specify their sex. The average age of the women was 26.9 years (*SD* = 11.6), and the average age of the men was 29.5 years (*SD* = 11.6). Moreover, 55.1% of the participants indicated that they were single, 38.3% reported being in an intimate relationship, and 6.6% indicated their relationship status as “other”. Additionally, single participants reported being single for an average of 5.3 years (*SD* = 7.3).

### 2.2. Materials

The study was conducted in Greek, run online, and designed using Google Forms. It consisted of seven parts. The first part measured perceived mate choice plurality, the second part measured choice overload, the third part measured self-esteem, the fourth part collected demographic information, the fifth part assessed regrets, the sixth part evaluated relationship satisfaction, and the seventh part measured willingness to stay in the relationship. The last three parts (five, six, seven) were completed only by participants who indicated that they were currently in a relationship or married. The order of presentation for each question in each part was randomized across participants.

To develop a measure of mate choice plurality, we followed these steps: We utilized AI tools (Chat GPT version 3.5 and Google Gemini) to generate items that could be used in a psychometric instrument to measure how many available mates people perceive they have. From the produced items, we selected 12 with the highest face validity. Based on this choice, we constructed a first version of the instrument and piloted it on a sample of 20 students. We conducted an internal consistency analysis using Cronbach’s alpha and dropped two items that reduced overall consistency. The final instrument consisted of 10 items (Appendix A), with a higher mean score indicating more perceived choices. The Cronbach’s alpha for this instrument was 0.89.

A similar procedure was followed to develop an instrument to measure choice overload in mate choice. We asked the AI tools to generate several items for developing a psychometric instrument that would measure difficulty in choosing between available mates deemed desirable. Through screening items for face validity and with a pilot study, we constructed an instrument that consisted of eight items (Appendix B), with a higher mean score indicating higher choice overload. The Cronbach’s alpha for this instrument was 0.92.

This procedure was also used to construct instruments to measure regrets in being in the current relationship, and willingness to stay in the current relationship. The regrets instrument consisted of four items (Appendix C), with a higher mean score indicating more regrets for being in the existing relationship. The Cronbach’s alpha for this instrument was 0.89. The willingness to stay in the relationship instrument consisted of four items (Appendix D), with a higher mean score indicating a higher willingness to stay in the current relationship. The Cronbach’s alpha for this instrument was 0.73. For all four instruments we developed, participants’ answers were recorded on the following scale: 1—Strongly disagree, 5—Strongly agree.

Additionally, we measured self-esteem using the Rosenberg Self-Esteem Scale, which consisted of 10 items with a higher score indicating higher self-esteem [28]. The Cronbach’s alpha for this scale was 0.89. For measuring relationship satisfaction, we employed a seven-item instrument developed [29], in which participants were asked to answer each item using a five-point scale ranging from “1” (low satisfaction) to “5” (high satisfaction). The Cronbach’s alpha for this scale was 0.90. With respect to demographic information, we recorded sex (man, woman, other), age, and relationship status (single—not currently having a stable partner, in an intimate relationship—currently having a stable partner, other). If participants indicated that they did not have an intimate partner, they were further asked to indicate for how many years they had been single.

### 2.3. Data Analysis

To identify the associations of interest on relationship status (Figure 1), we conducted a mediation analysis. More specifically, relationship status (single, in a relationship) was entered as the dependent variable, mate choice plurality as the independent variable, choice overload as the mediator, and sex, age, and self-esteem as covariates. To examine whether the associations were similar across sexes, we omitted the sex variable and repeated the analysis separately for male and female participants. We have also performed the mediation analysis, substituting relationship status with years single as the dependent variable.

Furthermore, to identify the associations of interest on the willingness to stay in the relationship (Figure 2), we conducted a serial mediation analysis. Specifically, willingness to stay in the relationship was entered as the dependent variable, mate choice plurality as the independent variable, regrets and relationship satisfaction as mediators, and sex, age, and self-esteem as covariates. As above, we omitted the sex variable and performed the analysis separately for men and women. In all cases, unstandardized indirect effects were calculated for each of the 10,000 bootstrapped samples, and the 95% confidence interval was computed by determining the indirect effects at the 2.5th and 97.5th percentiles. The analysis was performed using SPSS version 28 and the PROCESS version 4.2 macro.

## 3. Results

### 3.1. Preliminary Analysis

Initially, we calculated the percentage of participants who gave a mean score of four or more on the mate choice overload scale. Given the 1–5 scale used, this percentage provides insight into how many participants found it considerably difficult to choose between prospective mates. We found this number to be 10.6%. Furthermore, we performed a logistic regression to examine whether there were sex and age effects on this percentage. Specifically, the percentage was entered as the dependent variable, and sex and age were entered as independent variables. To control for any confounding effects, perceived mate choice plurality and self-esteem were entered as covariates. There was no significant main effect of age, but there was a significant main effect of sex [*X*^2^(1, N = 734) = 7.43, *p* = 0.006], with the odds ratio indicating that women were 1.8 times more likely than men to report experiencing high choice overload. Moreover, we correlated self-esteem and perceived mate choice plurality using Pearson’s product moment correlation. We found a significant positive correlation between the two variables [r(779) = 0.42, *p* < 0.001 (two-tailed)].

### 3.2. H1: Mate Choice Plurality Would Be Associated with Relationship Status

We will begin by discussing the results of the mediation analysis presented in Figure 1. Note that, as the dependent variable (relationship status) is categorical, the effects of mate choice plurality (c’), choice overload (b), and the indirect effect (a*b) are expressed in odds ratios in Table 1. On the other hand, the effect of mate choice plurality on choice overload is not expressed in odds ratios, as both variables are interval. Additionally, in mediation analysis where the dependent variable is years single, none of the results are in odds ratios (Table 2).

From Table 1, we can see that there was a significant direct effect of mate choice plurality on relationship status. In particular, more perceived choices were associated with a higher probability to be in an intimate relationship than single. More specifically, one unit increase in the perceived mate choice plurality was associated with a 1.62 times higher probability to be in an intimate relationship than single. The effect was present and of similar magnitude for both male and female participants. From Table 2, we can see that there was a significant effect of relationship status on years being single: More perceived mate choices were associated with fewer years being single. In particular, one unit increase in the mate choice plurality was associated with a 2.08-year decrease in the time being single. The effect was significant and of similar magnitude in both sexes.

### 3.3. H2: Mate Choice Plurality Would Be Associated with Choice Overload

From Table 1, we can observe a significant effect of mate choice plurality on choice overload, with an increase in perceived choices being associated with higher choice overload. Specifically, a one-unit increase in mate choice plurality was associated with a 0.32 unit increase in choice overload. This effect was significant and of similar magnitude for both male and female participants. Similar results were obtained when the analysis was performed on the duration of being single (Table 2).

### 3.4. H3: Choice Overload Would Be Associated with Relationship Status

From Table 1, we can observe that higher choice overload was associated with a higher incidence of singlehood. Specifically, a one-unit increase in choice overload was associated with a 26% [(1 − 0.64) × 100] decrease in the probability of being in an intimate relationship as opposed to being single. This effect was present for both men and women. On the other hand, as shown in Table 2, higher choice overload was associated with more years of being single; however, the effect was not significant.

### 3.5. H4: Mate Choice Plurality Would Be Indirectly Associated with Relationship Status through Choice Overload

From Table 1, we can see that there was a significant indirect effect of mate choice plurality on relationship status through choice overload. Specifically, a one-unit increase in mate choice plurality was associated with a 13% [(1 − 0.87) × 100] decrease in the probability of being in an intimate relationship rather than being single. This effect was significant and of almost identical magnitude for both male and female participants. On the other hand, this effect was not significant for the years being single (Table 2).

### 3.6. H5: Mate Choice Plurality Would Be Indirectly Associated with Willingness to Stay in the Relationship

We move on to discussing the results of the mediation analysis presented in Figure 2. Given that the dependent variable (willingness to continue the relationship) is interval, all results presented in Table 3 and Table 4 are not in odds ratios. From Table 3, we can observe that there was no significant direct effect of mate choice plurality on willingness to stay in the relationship. Additionally, there was no significant effect of mate choice plurality on relationship satisfaction, and no significant effect of regrets on willingness to stay in the relationship. However, there was a significant effect of mate choice plurality on regrets, with an increase in perceived choices being associated with more regrets. Furthermore, the regrets variable was significantly associated with relationship satisfaction, with more regrets being associated with lower relationship satisfaction. Additionally, higher relationship satisfaction was associated with higher willingness to stay in the relationship.

From Table 4, we can see that there was a significant indirect effect of mate choice plurality on willingness to stay in the relationship. More specifically, a one-unit increase in mate choice plurality was associated with a 0.06 unit decrease in willingness to stay in the relationship, which was associated with more regrets that, in turn, were associated with lower relationship satisfaction. We can further observe that this indirect association was significant only for male participants. For female participants, it was in the predicted direction, but it did not reach the level of significance.

## 4. Discussion

In the present research, we found that participants who perceived they had more dating options were less likely to be single, and spent fewer years single. On the other hand, more perceived mate choices were associated with more choice overload, which in turn, was associated with a higher likelihood of being single than in an intimate relationship. Furthermore, more perceived mate choices were associated with more regrets about being in the current relationship. These regrets were associated with lower relationship satisfaction, which in turn, was associated with lower willingness to stay in the relationship. Yet, this indirect effect was only significant for male participants.

People are selective when it comes to mating, so the more options they have, the more likely they are to find someone who meets their standards. Accordingly, and consistent with our original prediction, more perceived mate choices were associated with a decreased incidence of singlehood, and fewer years spent single. Still, having many choices is not always a good thing, as it can result in choice overload, with people facing difficulties deciding who is the best option for them, staying single as a consequence. Although the positive direct effect of perceived choices on relationship status was considerable—a one unit increase in it was associated with a more than 60% increase in the probability of being in an intimate relationship rather than single—the negative effect was also substantial—a one unit increase in it was associated with a 26% decrease in the probability of being in a relationship rather than single. The direct effect was larger, which means that people who have more choices are generally more likely to be in a relationship than people who have few choices. Studies on online dating are also consistent with the argument that choice overload could be an issue in the domain of mating [22], but the results of the present study are not confined to online dating.

In contrast to our original prediction, choice overload was not associated with years being single, and thus, there was no indirect effect of mate choice plurality on years being single through difficulties in choosing between mates. One possible interpretation of this finding is that mate choice plurality may hold people back from finding a mate, effectively increasing the probability of being single rather than in an intimate relationship, but most likely not for very long to have a noticeable impact on the years being single. This interpretation is consistent with the finding that the direct positive effect of mate choice plurality on relationship status is larger than the indirect negative effect. Nonetheless, more research is necessary to examine the effect of choice overload on years spent single.

In the present study, we examined perceived rather than actual mate choice plurality. The former is probably a better measure than the latter in testing our hypotheses. For instance, people who have many actual choices but perceive that they have only a few are unlikely to experience choice overload. Yet, perceived mate choice plurality may be subject to biases, such as people with high self-esteem overestimating the number of mates available to them. Consistent with this argument, we found a significant positive correlation of moderate size between perceived mate choice plurality and self-esteem. This correlation likely also captures the effect that having many choices boosts one’s self-esteem. By controlling for self-esteem, we ensured that the observed associations of mate choice plurality did not reflect self-esteem effects.

A bit more than one in ten participants in our sample reported high choice overload, indicating that this is not a negligible issue in the domain of mating. We also found that women were more likely than men to experience mate choice overload. As mate choice plurality was kept statistically constant, this finding suggests that for the same perceived number of available choices, women face more difficulties in deciding who is right for them. This result may reflect that, when it comes to mate choice, women are more selective than men [16], which makes it more difficult to decide between available options.

Additionally, mate choice plurality could also increase the probability of being single by making an existing relationship less satisfactory, thereby increasing one’s willingness to terminate it. This indirect effect was found only in men. One reason is that men can desire more variety in sexual partners [30]. Thus, more perceived mate choices may also be interpreted as more opportunities for casual sex. Accordingly, men may become more willing to end their current relationship in order to pursue such opportunities. We need to note that the observed effect was small. Yet, it reflects an average, and in some individual cases, it could be profound. For instance, men who have a great many mate choices, such as rock or movie stars, could easily become dissatisfied with their current relationship and become willing to terminate it. This “rock star effect” could possibly explain the high incidence of divorce among famous men. Additionally, opportunities for casual sex may not be the only reason why more perceived mate choices can lead to lower willingness to continue a relationship. For instance, it could be that men are more willing than women to engage in mate switching [31] if the opportunity for doing so arises. Thus, future studies need to investigate further the factors that lead to the observed effect.

Our study aimed to examine the direct and indirect effects of mate choice plurality on relationship status. Yet, mate choice plurality is itself predicted by several factors. For instance, personality could be such a factor: Extroverted people are more likely to have a wider social network [32] and thus, have more mating options than introverted people. On the other hand, people who live in a small village may have more limited mate options than people who live in a big city. Accordingly, future studies need to examine the factors that predict mate choice plurality and examine how they relate to singlehood. For example, it could be the case that on one hand, high extroversion enables people to find a suitable mate by increasing the probability that they meet someone who makes a good match, but on the other hand, it prevents them from finding a suitable mate by causing them choice overload. In the same vein, there may be factors that moderate choice overload such as relationship experience. For instance, it could be the case that individuals who are more experienced in the domain of mating are better at navigating between different mating options than less experienced ones, and thus, would be less susceptible to choice overload. Thus, future research needs to identify the moderators of choice overload.

Online dating refers to using internet-based websites or applications for entertainment, casual sex, or long-term romantic relationships [33]. In 1995, Match.com (accessed on 19 September 2023) launched as a public global online dating service. In the years that followed, the widespread access to the internet and the use of smartphones have resulted in the widespread use of online dating [34]. The COVID-19 pandemic has also resulted in more people looking for mates online [35]. The increasing use of online dating has triggered research attempting to understand its various aspects [36,37]. One issue with online dating is that it can potentially provide access to thousands or millions of possible mates [20], causing choice overload [21]. Given that online dating is on an increasing trajectory, we expect that in subsequent years it is going to become a more widespread issue, potentially affecting the prevalence of singlehood. This calls for more research investigating choice overload and ways to address it.

Choice overload appears in literature as a problem that holds people back from making choices [15]. Our results seem to be consistent with this interpretation: Assuming that singlehood is an undesirable state, mate choice plurality can hold people back from getting an intimate partner. Nevertheless, an alternative interpretation may also hold: Choice overload may prevent people from rushing into a relationship with the first person that seems to be a good choice, giving them more time to assess their options and perhaps make a wiser choice. In this interpretation, choice overload is not a brake that stops people from getting a mate, but a brake that reduces their speed of doing so, enabling them to make better choices. Some research on consumer behavior [6], and online dating [21] indicates that choice overload may lead to poorer choices, going against the latter interpretation. Additional research is necessary to assess the impact of choice overload on mating.

One limitation of the current research is that it was based on self-report instruments, which are subject to several biases such as people giving inaccurate responses. Another limitation is that we employed a non-probability sample, so our findings may not readily apply to the general population. This may not be a severe problem however. For instance, one study replicated 27 survey experiments originally conducted on nationally representative samples using online non-probability samples and found very high correspondence despite obvious differences in sample composition [38]. Still, further replication studies using more representative samples are necessary to increase our confidence that our findings apply to the general population. Moreover, the study was confined to the Greek cultural context, and its results may not apply to other cultural settings. In non-Western societies, where mating is more strictly controlled, and parents have a considerable role in determining their children’s mate choices [39], mate choice plurality could be a more important issue for mate-seekers, but choice overload may not be. On the other hand, in societies where marriages are arranged, choice overload could be a problem for parents when exercising mate choice. Cross-cultural research is necessary for examining how cultural factors affect the impact of mate choice plurality and choice overload on relationship status. Furthermore, as discussed above, another limitation of the current work is that it did not examine the factors that possibly predict mate choice plurality, and the factors that may moderate choice overload. Additionally, in the current study, we did not record participants’ sexual orientation. However, given that most individuals are heterosexual, it is reasonable to assume that the majority of participants were heterosexual. We do not have any reasons to believe that our conclusions do not hold for non-heterosexual individuals; nevertheless, future studies need to attempt to replicate our findings in non-heterosexual samples.

The current research was designed to address the question of whether more choices are better for finding an intimate partner. Our results indicate that the answer is generally yes, but too many choices can hold people back from finding the right one. Future studies are necessary in order to better understand how mate choice plurality and choice overload relate to relationship status.

## Figures and Tables

**Figure 1 behavsci-14-00703-f001:**
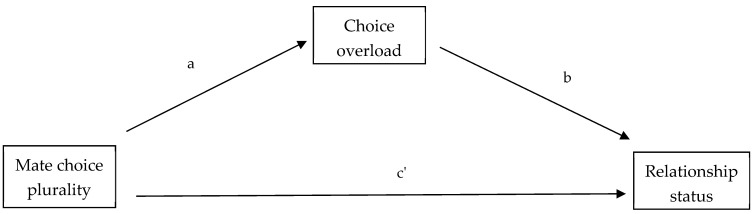
The figure above depicts the direct and indirect effect of mate choice plurality on relationship status.

**Figure 2 behavsci-14-00703-f002:**
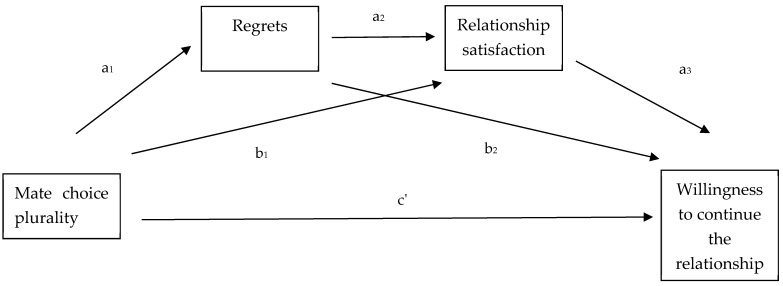
The figure above depicts the different pathways that mate choice plurality can affect one’s willingness to continue the current intimate relationship.

**Table 1 behavsci-14-00703-t001:** Direct and indirect effects of mate choice plurality on relationship status.

	Mate Choice Plurality on Relationship Status (c’)	Mate Choice Plurality on Choice Overload (a)	Choice Overload on Relationship Status (b)	Mate Choice Plurality * Choice Overload (a*b)
Total	1.62 ** (1.23–2.14)	0.32 ** (0.20–0.44)	0.64 ** (0.51–0.80)	0.87 ** (0.77–0.94)
Women	1.68 ** (1.29–2.19)	0.34 ** (0.22–0.46)	0.63 ** (0.51–0.77)	0.82 ** (0.72–0.95)
Men	1.50 * (1.07–2.012)	0.28 ** (0.15–0.43)	0.65 * (0.49–0.86)	0.88 ** (0.73–0.98)

* significant at 0.05; ** significant at 0.001; Note. With the exception of (a), all effects are expressed in odds ratios where the reference category is ‘single’.

**Table 2 behavsci-14-00703-t002:** Direct and indirect effects of mate choice plurality on years single.

	Mate Choice Plurality on Years Single (c’)	Mate Choice Plurality on Choice Overload (a)	Choice Overload on Years Single (b)	Mate Choice Plurality * Choice Overload (a*b)
Total	−2.08 ** (−3.26–−0.90)	0.40 ** (0.24–0.56)	0.61 (−0.32–1.54)	0.25 (−0.12–0.68)
Women	−2.21 ** (−3.83–−0.60)	0.45 ** (0.24–0.66)	0.87 (−0.42–1.16)	0.39 (−0.16–1.11)
Men	−2.03 * (−3.81–−0.24)	0.34 ** (0.09–0.58)	0.22 (−1.15–1.58)	0.07 (−0.43–0.68)

* significant at 0.05; ** significant at 0.001.

**Table 3 behavsci-14-00703-t003:** Serial mediation analysis results: direct effects.

	Mate Choice Plurality on Willingness to Continue the Relationship (c’)	Mate Choice Plurality on Relationship Satisfaction (b_1_)	Mate Choice Plurality on Regrets (a_1_)	Regrets on Relationship Satisfaction (a_2_)	Regrets on Willingness to Continue the Relationship (b_2_)	Relationship Satisfaction on Willingness to Continue the Relationship (a_3_)
Total	0.06 (−0.10–0.21)	0.06 (−0.04–0.16)	0.28 ** (0.07–0.49)	−0.47 ** (−0.54–−0.40)	0.04 (−0.12–0.20)	0.45 ** (0.20–0.70)
Women	0.05 (−0.10–0.21)	0.05 (−0.04–0.15)	0.24 ** (0.03–0.45)	0.48 ** (−0.54–−0.41)	0.01 (−0.15–0.16)	0.43 ** (0.19–0.67)
Men	0.03 (−0.23–0.29)	0.07 (−0.11–0.26)	0.41 ** (0.09–0.73)	0.45 ** (−0.59–−0.31)	0.10 (−0.17–0.36)	0.46 ** (0.08–0.83)

** significant at 0.001.

**Table 4 behavsci-14-00703-t004:** Serial mediation analysis results: indirect effects.

	Mate Choice Plurality * Regrets (a_1_*b_2_)	Mate Choice Plurality * Relationship Satisfaction (b_1_*a_3_)	Mate Choice Plurality * Regrets * Relationship Satisfaction (a_1_*a_2_*a_3_)
Total	0.01 (−0.04–0.08)	0.03 (−0.02–0.08)	−0.06 ** (−0.13–−0.01)
Women	0.00 (−0.04–0.06)	0.02 (−0.02–0.07)	−0.05 (−0.15–0.02)
Men	0.04 (−0.08–0.20)	0.03 (−0.05–0.13)	−0.09 ** (−0.23–−0.01)

** significant at 0.001.

## Data Availability

All data are available from the first author on request.

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
