# Peer review of "Mate Choice Plurality, Choice Overload, and Singlehood: Are More Options Always Better?"

_behavsci, 2024, doi:10.3390/bs14080703_

Round 1

Reviewer 1 Report

Comments and Suggestions for Authors

Dear Authors,

I have read with pleasure the present article about the interconnection between mate choice plurality, choice overload, and singlehood in sample of 804 Greek-speaking participants. The topic is interesting, not only because it solves several puzzling topics related to human relationships, but it can only serve for further research.

Down below, there are some specific comments reported related to each part of the paper:

Abstract

-        Lack of clarity

-        Not enough to summarize the findings and the protocol of the article

-        More statements than assumptions in the studied topic

-        Results are copy-pasted from the discussion.

Introduction

-        Lack of neutrality and overgeneralizing in some statements. à while listing contradicting arguments, some are more referenced and more followed.

-        Weak organization of the paragraphs à lots of back-and-forth

-        Hypothesis distributed all over the place à hard to follow at some point.

Material and Methods

-        Clear and organized compared to other parts.

Further comments

-        Absence of homogeneity in general

-        Complex syntax and long sentences sometimes make it harder to read and follow.

-        Intriguing topic, but the arrangement and structuring of the article makes it quite challenging for the reader to read it.

Author Response

Dear Authors,

I have read with pleasure the present article about the interconnection between mate choice plurality, choice overload, and singlehood in sample of 804 Greek-speaking participants. The topic is interesting, not only because it solves several puzzling topics related to human relationships, but it can only serve for further research.

We would like to thank you very much for your feedback and for your kind words about our work. Yes, we agree with you that this is an interesting topic, and we hope that our paper will trigger further research in the area. Please see below how we have addressed your concerns.

Down below, there are some specific comments reported related to each part of the paper:

Abstract

-        Lack of clarity

-        Not enough to summarize the findings and the protocol of the article

-        More statements than assumptions in the studied topic

-        Results are copy-pasted from the discussion.

Following your suggestion, we have reworked the Abstract in order to improve clarity, reduce repetition, and provide more information about the methodology of our study.

Introduction

-        Lack of neutrality and overgeneralizing in some statements. à while listing contradicting arguments, some are more referenced and more followed.

-        Weak organization of the paragraphs à lots of back-and-forth

-        Hypothesis distributed all over the place à hard to follow at some point.

Following your suggestion, in order to increase the readability of the introduction, we have added a summary of our arguments at the end of the first paragraph of the introduction. Moreover, we agree with you that keeping track of all the hypotheses may be an issue thus, we have summarized all the hypotheses in the 1.2 The current study section. We have also worked on the introduction to improve readability, but please let us know if there are any specific sentences or paragraphs that you think they require more work.

Material and Methods

-        Clear and organized compared to other parts.

Further comments

-        Absence of homogeneity in general

-        Complex syntax and long sentences sometimes make it harder to read and follow.

-        Intriguing topic, but the arrangement and structuring of the article makes it quite challenging for the reader to read it.

We worked on the manuscript to improve readability. We would like to say however, that in the current study, we aimed to explore indirect effects, and describing them involves using long sentences. Moreover, we tested several hypotheses, and we assessed the effects of several different variable, which inevitably has an effect on the homogeneity of the manuscript. However, please let us know if you think there are specific sentences or sections that need further work.

We would like to thank you very much once more for considering our work.

Reviewer 2 Report

Comments and Suggestions for Authors

The manuscript presents an exploration of the relationship between the perception of having a wide choice of romantic partners and various circumstances such as being single or in a relationship, and relationship satisfaction. The article is well structured and well written. However, two significant issues have been identified that impact the quality of the manuscript.

Firstly, the Authors briefly touch upon the topic of online dating to assert the breadth of their approach. However, given the current interest in the increasing widespread of online dating and its potential effects, a more extensive engagement with this theme and its relevant literature is warranted. Consequently, in its present form the article show a noticeable absence of theoretical reflection and literature review .

Secondly, the methodological section, particularly regarding data collection, is weak. The Authors describe utilizing a non-probabilistic sampling method (par 2.1), raising concerns about the generalizability of their results. Without a probabilistic sample, the statistical analyses presented in the article lose their value, rendering any meaningful conclusions untenable. Unfortunately, this issue is neither addressed nor problematized in the manuscript.

It is recommended that the Authors address these issues before considering the manuscript for publication. Specifically, they should expand their discussion to include a thorough examination of the role of online dating and its implications for their study. Additionally, they must critically discuss the limitations of their non-probabilistic sampling method and its impact on the validity and generalizability of their findings.

Author Response

The manuscript presents an exploration of the relationship between the perception of having a wide choice of romantic partners and various circumstances such as being single or in a relationship, and relationship satisfaction. The article is well structured and well written. However, two significant issues have been identified that impact the quality of the manuscript.

We would like to thank you very much for considering our work and for providing us with valuable feedback. Please see below how we have addressed all your concerns and recommendations.

Firstly, the Authors briefly touch upon the topic of online dating to assert the breadth of their approach. However, given the current interest in the increasing widespread of online dating and its potential effects, a more extensive engagement with this theme and its relevant literature is warranted. Consequently, in its present form the article show a noticeable absence of theoretical reflection and literature review.

We agree with your argument that more discussion of the relevant literature is in order. Accordingly, we have added paragraph to discuss online dating further and to provide additional references (Discussion, seventh paragraph, line 464). We would like to say that, as this is not a paper about online dating, we feel that it would not be appropriate to expand on this further.

Secondly, the methodological section, particularly regarding data collection, is weak. The Authors describe utilizing a non-probabilistic sampling method (par 2.1), raising concerns about the generalizability of their results. Without a probabilistic sample, the statistical analyses presented in the article lose their value, rendering any meaningful conclusions untenable. Unfortunately, this issue is neither addressed nor problematized in the manuscript.

Yes, you make a good point- we employed a non-probability sample so our findings may not readily generalize to the population, a limitation that needs to be discussed in the text. Following your suggestion, we have now expanded the limitations section, to make discuss this issue in detail (Discussion, second paragraph from the end, line 488).

It is recommended that the Authors address these issues before considering the manuscript for publication. Specifically, they should expand their discussion to include a thorough examination of the role of online dating and its implications for their study. Additionally, they must critically discuss the limitations of their non-probabilistic sampling method and its impact on the validity and generalizability of their findings.

We would like to thank you very much once more for considering our work.

Reviewer 3 Report

Comments and Suggestions for Authors

The paper covers an understudied topic potentially abundant with trajectories for academics. The abstract appears to have a slightly unclear sentence in lines 19~20 (repeated again in lines 393~395), i.e.: "We found that more perceived mate choices were associated with a lower likelihood of being single compared to being in an intimate relationship." In other words, if someone had more choices for partners, then that person would have a greater chance of simply ending the status of that person's singlehood, but that chance would involve a non-intimate (i.e., superficial) relationship? (It seems that lines 440~441 ("Additionally, mate choice plurality could also increase the probability of being single by making an existing relationship less satisfactory, thereby increasing one’s willingness to terminate it") serves to explain lines 19~20 and lines 393~395, but perhaps the authors can make this explanation clearer.

In the introduction of lines 1~66, the authors should explore a more interdisciplinary review of literature (one has the feeling that the authors considered literature that merely confirmed general relationship trends, but not the nuances or circumstances surrounding those trends). In defense of the authors, the authors then say (in lines 109~111) that the scholarly literature does not have much to say on the thesis of this study, but the authors ought to still consider indirect or tangentially related materials in the academic world.

In lines 147~155, despite the assertions of the authors, we still found ourselves wondering about people with low self-esteem---in other words, perhaps the authors could try to offer some provisional conjectures regarding the notion of whether or not individuals with low self-esteem ought to have a deeper inclination towards having a satisfied feeling with a relationship.

On the other hand, the authors should consider reasons outside of sexual satisfaction (cf. lines 442~443) for why men (in phenomena of mate choice plurality) seem more inclined to terminate relationships than women do. Perhaps the authors might expand on the conclusion by considering non-heterosexual romantic relationships and whether or not the dynamics of those relationships resemble or diverge from the circumstances of heterosexual relationships, which seem to have formed the core foundation of the conclusions reached in the study.

Author Response

We would like to thank you very much for considering our work and for your feedback that enabled us to improve our manuscript. Please see below how we have addressed all your concerns and recommendations.

-The paper covers an understudied topic potentially abundant with trajectories for academics. The abstract appears to have a slightly unclear sentence in lines 19~20 (repeated again in lines 393~395), i.e.: "We found that more perceived mate choices were associated with a lower likelihood of being single compared to being in an intimate relationship." In other words, if someone had more choices for partners, then that person would have a greater chance of simply ending the status of that person's singlehood, but that chance would involve a non-intimate (i.e., superficial) relationship? (It seems that lines 440~441 ("Additionally, mate choice plurality could also increase the probability of being single by making an existing relationship less satisfactory, thereby increasing one’s willingness to terminate it") serves to explain lines 19~20 and lines 393~395, but perhaps the authors can make this explanation clearer.

Following your feedback, we realized that this sentence may be indeed unclear so we have rephrased it as follows:

We found that participants who perceived they had a wider range of potential romantic partners reported a lower likelihood of being single. (Abstract)

We have also rephrased the first sentence of the Discussion section as follows:

In the present research, we found that participants who perceived they had more dating options were less likely to be single, and spent fewer years single.

-In the introduction of lines 1~66, the authors should explore a more interdisciplinary review of literature (one has the feeling that the authors considered literature that merely confirmed general relationship trends, but not the nuances or circumstances surrounding those trends). In defense of the authors, the authors then say (in lines 109~111) that the scholarly literature does not have much to say on the thesis of this study, but the authors ought to still consider indirect or tangentially related materials in the academic world.

We aimed for our literature review to be concise and focused on the objectives of the study. In particular, we discuss the literature on choice overload and singlehood, and we specifically state that, to the best of our knowledge, this is the first study that has attempted to examine how the two relate. We do not disagree that there are other studies broadly relevant here. Nevertheless, the fields of mating and choice are extensive, so we do not think it would be fruitful to expand the literature review with studies that do not directly relate to the objectives of the current work.

-In lines 147~155, despite the assertions of the authors, we still found ourselves wondering about people with low self-esteem---in other words, perhaps the authors could try to offer some provisional conjectures regarding the notion of whether or not individuals with low self-esteem ought to have a deeper inclination towards having a satisfied feeling with a relationship.

That’s a good point. Following your suggestion, we have expanded the last paragraph of the ‘1.2 The current study section’ to discuss the role of self-esteem on relationship satisfaction.

-On the other hand, the authors should consider reasons outside of sexual satisfaction (cf. lines 442~443) for why men (in phenomena of mate choice plurality) seem more inclined to terminate relationships than women do. Perhaps the authors might expand on the conclusion by considering non-heterosexual romantic relationships and whether or not the dynamics of those relationships resemble or diverge from the circumstances of heterosexual relationships, which seem to have formed the core foundation of the conclusions reached in the study.

Yes, following your suggestion, we now state specifically that there may be factors other than casual sex that can account for the observed effect (Discussion, end of sixth paragraph). Moreover, following your suggestion, we now discuss whether our findings apply to non-heterosexual populations (Discussion, end of second paragraph from the end).

Round 2

Reviewer 2 Report

Comments and Suggestions for Authors

The authors have made some adjustments based on the initial feedback, particularly by adding a some lines addressing the literature on online dating and acknowledging the non-probabilistic nature of their sample. However, there are still significant concerns that need to be addressed.

Firstly, while the addition of lines about online dating literature is a step in the right direction, I think that this discussion remains minimal and should be incorporated more comprehensively at the beginning of the article.

Secondly, and more critically, the issue with the non-probabilistic sample remains inadequately addressed. The authors have only added a brief statement at the end of the article noting the non-probabilistic nature of the sample, accompanied by a few lines suggesting that this does not pose a significant problem.

The manuscript is structured around hypothesis testing, data collection, and statistical analysis. For such an approach, a probabilistic sample is essential. It is not methodologically sound to discuss statistical significance of results without a probabilistic sample. To address this, the authors should either collect data using a probabilistic sampling method , or reframe the study as a preliminary exploration of hypotheses, clearly stating at the beginning of the article that the findings are not statistically significant due to the non-probabilistic sample, ensuring readers understand the exploratory nature of the research.

Author Response

The authors have made some adjustments based on the initial feedback, particularly by adding a some lines addressing the literature on online dating and acknowledging the non-probabilistic nature of their sample. However, there are still significant concerns that need to be addressed.

-We would like to thank you very much for considering our work. We fully understand your concerns but would prefer not to change our manuscript in ways that we do not feel comfortable with—for instance, by presenting significant results as non-significant or devoting a large part of the manuscript to discussing literature that is not related to our study. Please see more details below.

Firstly, while the addition of lines about online dating literature is a step in the right direction, I think that this discussion remains minimal and should be incorporated more comprehensively at the beginning of the article.

-The current paper is not about online dating (although it may be relevant for people working in that area). Accordingly, it would be inappropriate and misleading to discuss online dating extensively in the introduction.

Secondly, and more critically, the issue with the non-probabilistic sample remains inadequately addressed. The authors have only added a brief statement at the end of the article noting the non-probabilistic nature of the sample, accompanied by a few lines suggesting that this does not pose a significant problem.

The manuscript is structured around hypothesis testing, data collection, and statistical analysis. For such an approach, a probabilistic sample is essential. It is not methodologically sound to discuss statistical significance of results without a probabilistic sample. To address this, the authors should either collect data using a probabilistic sampling method , or reframe the study as a preliminary exploration of hypotheses, clearly stating at the beginning of the article that the findings are not statistically significant due to the non-probabilistic sample, ensuring readers understand the exploratory nature of the research.

-You suggest that we treat our significant results as not significant on the grounds that we employed a non-probability sample. This would be statistically inappropriate, as significance is not related to the sampling procedure. The sampling procedure impacts the generalizability of our results to the broader population, a limitation that we have fully acknowledged in the text. However, please note that we have employed a large and diverse sample of participants, so we consider it highly unlikely that our results would differ in a probability sample.

Reviewer 3 Report

Comments and Suggestions for Authors

The authors have fulfilled the spirit of the recommendations offered in the first round of revisions. In the literature review section, we understand the reluctance of the authors to explore indirectly related literature.

Author Response

Thank you so much once more for your feedback!

Round 3

Reviewer 2 Report

Comments and Suggestions for Authors

Dear Authors, I would like to point out that in my opinion the issue regarding sampling procedures still appears unresolved. Specifically, the assertion that a non-probability sample can be used as if it were a probability sample remains an hypothesis to be supported within your article. This aspect is crucial for the study's conclusions and requires further methodological justification and evidence.

Author Response

Please see our original response.